# Integrated Transcriptomic, Metabolomic, and Physiological Analyses Reveal New Insights into Fragrance Formation in the Heartwood of *Phoebe hui*

**DOI:** 10.3390/ijms232214044

**Published:** 2022-11-14

**Authors:** Hanbo Yang, Wenna An, Fang Wang, Yunjie Gu, Hongying Guo, Yongze Jiang, Jian Peng, Minhao Liu, Lianghua Chen, Fan Zhang, Peng Zhu, Xiong Huang, Xueqin Wan

**Affiliations:** 1Forestry Ecological Engineering in the Upper Reaches of the Yangtze River Key Laboratory of Sichuan Province, National Forestry and Grassland Administration Key Laboratory of Forest Resource Conservation and Ecological Safety on the Upper Reaches of the Yangtze River, Rainy Area of West China Plantation Ecosystem Permanent Scientific Research Base, Institute of Ecology and Forestry, Chengdu 611130, China; 2Sichuan Key Laboratory of Ecological Restoration and Conservation for Forest and Wetland, Sichuan Academy of Forestry, Chengdu 610081, China; 3Sichuan Academy of Grassland Sciences, Chengdu 611731, China

**Keywords:** *Phoebe hui*, heartwood, fragrance, volatile metabolites, transcriptome, physiological, terpenoid

## Abstract

*Phoebe hui* is an extremely valuable tree that is the main source of the fragrant golden-thread nanmu wood. Although the fragrance of wood has been investigated in several trees, the potential substances and gene regulation mechanisms that are involved in fragrance formation are poorly understood. Here, three radial tissues, sapwood (SW), heartwood (HW), and the transition zone (TZ) in between them, were compared via integrative physiological, volatile-metabolomic, and transcriptomic analyses to identify the key metabolites and regulatory mechanisms involved in fragrance formation. During heartwood formation, gradual starch grain loss was accompanied by the deposition of lipids and extractives in the cell lumen. Extracts of terpenoids were synthesized and accumulated in the heartwood, including monoterpenoids (limonene and p-cymene) and sesquiterpenes (cubebene and guaiadiene); these were identified as being closely related to the special fragrance of the wood. Additionally, the expression of transcripts showed that the genes related to primary metabolism were specifically upregulated in the SW, whereas genes annotated in terpenoid biosynthesis were specifically upregulated in the HW. Therefore, we speculated that terpenoid biosynthesis occurs in situ in the HW via the HW formation model of Type-III (*Santalum*) using the precursors that were produced by primary metabolism in the SW. The expression levels of transcription factors (e.g., MYB, WRKY, and C2H2) acted as the major regulatory factors in the synthesis of terpenoids. Our results explain the special fragrance in *P. hui* and broaden the current knowledge of the regulatory mechanisms of fragrance formation. This work provides a framework for future research that is focused on improving wood quality and value.

## 1. Introduction

For humans, wood products have played a central role in the cultural and economic development of society [1]. Golden-thread nanmu is a well-known, highly valuable timber industry product because of its attractive golden-yellow color, distinctive fragrance, and durability [2]. *Phoebe hui* Cheng ex Yang, which belongs to the *Phoebe* genus of the Lauraceae family, is highly valuable because of its attractive visible golden “threads” and unique fragrance. This species is one of the main sources of golden-thread nanmu wood. With its high-quality wood, *P. hui is* widely used in upscale buildings, furniture, sculptures, and landscaping [3]. Jiao et al. [4] reported that *P. hui* and *P. zhennan* are most likely the main species of nanmu used in the Forbidden City in Beijing, China (the largest wooden palace complex in the world). Furthermore, *P. hui* is a source of essential oils containing volatile organic compounds (VOCs) with high economic value in the spice and perfume industries [5]. Therefore, exploration of the fragrance-related volatile metabolites and the underlying mechanism of their biosynthesis in *P. hui* can enhance its production and overall quality in future breeding programs.

The wood of many tree species can be separated into sapwood and heartwood [1]. In comparison with sapwood, heartwood has a high value because of its special fragrance, beautiful color, and durability [6]. As the tree grows, sapwood is converted to heartwood through complex physiological and biochemical processes using reserve materials such as starch, which are consumed in the narrow transition zone (the intermediate developmental stage between sapwood and heartwood) [7,8,9]. Previous studies have reported that oxidase activities were also involved in the formation of heartwood [10]. The gradual loss of reserve materials is accompanied by the deposition of extractives (secondary metabolites) into the cell lumens of the heartwood [11]. These extractives confer the characteristics of fragrance, color, and durability to the heartwood [12]. For example, polyphenolic compounds, including anthocyanins, flavonoids, and flavones, are important common extractives that are responsible for the formation of red heartwood in *Cunninghamia lanceolate* [13]. Terpenoids also commonly accumulate in heartwood and are closely related to essential oil and fragrance production [1]. In *Taiwania cryptomerioides*, terpenoids were one major component of the heartwood extractives that corresponded to its special fragrance [14]. The fragrance-defining sesquiterpenoids were identified as the main components of sandalwood oil in *Santalum album* heartwood [15]. Xie et al. [16] demonstrated that monoterpenoids, sesquiterpenoids, and diterpenoids are present in the essential oil of *Cryptomeria fortunei*. Ghadiriasli et al. [17] identified the monoterpenes and sesquiterpenes that gave the wood of *Pinus cembra* a special odor. Schreiner et al. [18] showed that terpenoids were the main source of the special odor of *Pinus sylvestris* wood. Aromatic and aldehyde ketones were identified as the main components of odor in paper mulberry wood [19]. Furthermore, several studies have reported that *Phoebe* trees emit fragrances that are dominated by sesquiterpenoids [20,21,22,23].

Previous studies have demonstrated that heartwood formation and the biosynthesis of secondary metabolites are mainly genetically determined [19,24,25]. The extractives (secondary metabolites) are the final products of metabolism under complex gene regulatory networks. Several studies have reported that many upregulated pathway genes and transcription factors in the transition zone are closely related to the biosynthesis of terpenoids and flavonoids [24,25]. Terpenoids are synthesized by terpene synthases (TPSs) via prenyl diphosphates: geranyl-PP (GPP) for monoterpenes, farnesyl-PP (FPP) for sesquiterpenes, and geranylgeranyl-PP (GGPP) for diterpenes in plants [26]. Yeh et al. [24] found that the highly expressed TPS-encoding, GGPS-encoding, and FPS-encoding genes in the mevalonate pathway (MVA pathway) and methylerythritol 4-phosphate pathway (MEP pathway) in the transition zone contributed to the gradual increase in the accumulation of terpenoids from the transition zone toward the heartwood in *T. cryptomerioides*. In contrast, the biosynthesis of sesquiterpenoids of *S. album* occurs in the heartwood within unique living ray parenchyma cells under the regulation of FPS- and TPS-encoding transcripts [1,15]. Moniodis et al. [27] cloned the full-length cDNA and characterized the enzyme function of SspiTPS4 in *S. spicatum* heartwood xylem as a multiproduct sesquisabiene B synthase. Based on the above findings, Celedon and Bohlmann [1] discussed a new model of heartwood formation (Type-III: *Santalum*-Type) and suggested that the biosynthesis of terpenoids occurred in heartwood. Therefore, whether the heartwood-specific expression of genes regulates terpenoid synthesis remains to be further studied. In addition, transcription factors (TFs) are involved in the regulation of multiple key genes in terpenoid metabolism-related gene clusters [28]. Currently, six TF families (AP2/ERF, bHLH, MYB, NAC, WRKY, and bZIP) have been reported to be involved in the biosynthesis of terpenoids [29]. These findings suggest that the biosynthesis and deposition of fragrance-related metabolites in heartwood are actively regulated and represent a continuous stage of development in tree species; however, several questions remain regarding the formation and regulatory mechanisms of the fragrance-related secondary metabolites in *P. hui* heartwood, although the significance of the volatile metabolites and related genes have been studied in some woody plants. Given this gap in knowledge, we conducted physiological, metabolomic, and transcriptomic analyses of different radial tissues in the *P. hui* trunk to address the following questions: (1) what volatile metabolites are responsible for the fragrance in heartwood and (2) which candidate genes regulate the biosynthesis of fragrance-related volatile metabolites?

## 2. Results

### 2.1. Radial Distribution of Bioactive Compounds and Enzyme Activities

During heartwood formation, extractives (e.g., phenols and flavonoids) are synthesized using non-structural polysaccharides (e.g., starch) in the parenchyma cells through the catalysis of enzymes and are deposited in the heartwood. The contents of the total phenols and flavonoids in sapwood (SW), the transition zone (TZ), and heartwood (HW) were determined in order to explore the radial distribution of phenols across the stem of the tree (Figure 1). The results of the ANOVA showed that the total phenol content was significantly different (*F* = 4.920, *p* = 0.040) among SW, TZ, and HW, but there was no significant difference (*F* = 1.033, *p* = 0.382) in flavonoid content. The total phenol content in HW was significantly higher than that in SW. There were significant differences in the polyphenoloxidase (PPO) (*F* = 6.686, *p* = 0.009) and peroxidase (POD) (*F* = 7.601, *p* = 0.018) activities among the three tissues. The activity of POD in TZ was significantly higher than those in SW and HW, but the activity of PPO in SW was significantly higher than that in HW (Figure 1). Lipids, which were stained yellow with Sudan IV, were always observed in ray parenchyma cells, and the accumulations in HW and TZ were higher than that in SW, whereas extractives were deposited in HW and TZ (Figure 2A–C). Starch mainly accumulated in SW and TZ, and the amount of starch decreased from SW to HW (Figure 2D–F). The heartwood, where the extractives were deposited, was free of starch.

### 2.2. The Detection of Volatile Metabolites Related to Fragrance

To compare the differences in the composition of volatile metabolites in *P. hui* woods from the SW, TZ, and HW, we analyzed the three tissues using GC-MS. Twenty-four of 49 volatile metabolites in all samples were annotated to the public database and classified into 16 subclasses, including three sesquiterpenoids, three monoterpenoids, three alkanes, two benzoic acids and its derivatives, and two carbonyl compounds (Appendix A, Appendix A). A principal component analysis (PCA) of nine samples revealed that the first principal component (PC1) and the second principal component (PC2) accounted for 80.9% of the total variation. The volatile metabolites were clearly separated into three groups in the PCA (Appendix A), thereby suggesting differential accumulation among the three tissues, particularly for HW vs. SW and TZ vs. SW. To identify the accumulated volatile metabolites associated with the fragrance of *P. hui* heartwood, an orthogonal partial least squares-discriminant analysis (OPLS-DA) was performed and volcano plots were constructed (Appendix A). Based on a VIP ≥ 1, 24, 23, and 27, differentially accumulated metabolites (DAMs) were screened from the comparison groups of HW vs. SW, HW vs. TZ, and TZ vs. SW, respectively. Based on an FC > 1 and a *p*-value < 0.05, ten, zero, and two DAMs with an increased accumulation in HW were screened from the three comparison groups (HW vs. SW, HW vs. TZ, and TZ vs. SW, respectively). Finally, ten significant DAMs (VIP ≥ 1, *p*-value < 0.05) with an increased accumulation of volatile metabolites (FC > 1) in HW, including two monoterpenoids (limonene and p-cymene), two sesquiterpenoids (cubebene and guaiadiene), and one isoxazole, were chosen as potential metabolite markers that explained the fragrance in *P. hui* heartwood (Appendix A, Figure 3).

### 2.3. Transcriptome Analysis and DEGs among HW, TZ, and SW

The RNA yield decreased from SW to HW (SW > TZ > HW) (Appendix A). The high correlation coefficient indicates a strong linear relationship between biological duplications (Appendix A). A total of 18,021,025–24,142,378 clean reads of the nine RNA-seq libraries of SW, TZ, and HW were mapped to 193 locations in the reference genome of *Phoebe bournei* with a high mapping rate (67.79–87.30%) (Appendix A), indicating that the selected reference genome was appropriate. A detailed analysis of the number of genes identified 7401 novel genes, of which 4616 were successfully annotated, thus enriching the genomic information related to *P. hui* wood formation (Appendix A).

A total of 389 DEGs were identified among HW, TZ, and SW under the filtering criteria of a |log2(fold change)| > 1 and FDR < 0.05 (Appendix A). Of these DEGs, 250 DEGs (210 downregulated and 40 upregulated), 136 DEGs (129 downregulated and 7 upregulated), and 3 DEGs (1 upregulated and 2 downregulated) were identified in HW vs. SW, TZ vs. SW, and HW vs. TZ, respectively (Figure 4A–C). A total of 150 DEGs (35 upregulated and 115 downregulated) were found exclusively in HW vs. SW, which would be related to the biosynthesis of volatile metabolites in the heartwood (Figure 4D). The 150 DEGs were enriched in three main functional groups (biological processes, cellular components, and molecular functions) via GO term enrichment analysis (Figure 4E). “Cell process”, “membrane”, and “binding” were the largest groups in the categories of biological processes, cellular components, and molecular functions, respectively. Phenylpropanoid biosynthesis and amino sugar and nucleotide sugar metabolism were the dominant enriched pathways based on the KEGG pathway analysis (Figure 4F). Most unigenes related to pentose and glucuronate intercoversions, amino sugar and nucleotide sugar metabolism, and glyoxylate and dicarboxylate metabolism were upregulated in the SW (Figure 4G). Some DEGs were enriched in the pathways of fragrance-related metabolites (e.g., monoterpenoid biosynthesis and diterpenoid biosynthesis). We further verified the enrichment pathway of these metabolites using gene set enrichment analysis (GSEA) for the whole transcriptome. Unigenes annotated in diterpenoid biosynthesis were significantly (FDR q value = 0.04) enriched and specifically upregulated in HW (Figure 4H), and starch and sucrose metabolism and amino sugar and nucleotide sugar metabolism were significantly enriched and specifically upregulated in SW (Appendix A). Furthermore, the results of RT-qPCR showed that the expression patterns of the selected genes were consistent with the transcriptome data (Appendix A), which suggests that the results of our RNA-seq analysis were accurate.

### 2.4. Analysis of the Key Pathways Involved in Terpenoid Biosynthesis in P. hui Heartwood

Primary metabolism provides the precursors for the biosynthesis of terpenoids (Figure 5). Pectinesterase converts poly(1,4-D-gulacturonide into poly(1,4-D-gulacturonide) (n), and polygalacturonase then converts poly(1,4-D-gulacturonide) (n) into D-galacturonate in a phentose and glucuronate interconversion. We observed a 2.1- and 1.2-fold upregulation of pectinesterase-encoding transcripts in HW and TZ, respectively, compared with that in SW. In contrast, a 2.5- and 2.0-fold upregulation of polygalacturonase-encoding transcripts was detected in SW compared with those in HW and TZ, respectively. In the downstream pathway of amino sugar and nucleotide sugar metabolism, UDP-arabinose 4-epimerase (UXE) converts L-arabinose into UDP-glucose. We observed a 2.5- and 2.0-fold upregulation of polygalacturonase and UXE, uxe-encoding transcripts in SW compared with those in HW and TZ, respectively. Starch and sucrose metabolism-related transcripts (GN1_2_3 and AMY, amyA, malS) were also upregulated in the SW. Upstream of monoterpenoid, diterpenoid, sesquiterpenoid, and triterpenoid biosynthesis is the terpenoid backbone biosynthesis pathway that provides geranyl-PP (GPP), geranylgeranyl-PP (GGPP), and farnesyl-PP (FPP) for terpene biosynthesis. Further downstream, transcripts encoding (-)-alpha-terpineol synthase, which converts geranyl-PP to terpineol in monoterpenoid biosynthesis, were upregulated in SW; however, we detected an increased accumulation of limonene (a monoterpenoid) in the HW and TZ. In contrast, in the diterpenoid biosynthesis pathway, momilactone-A synthase (MAS)-related transcripts were 2.0- and 1.1-fold upregulated in the HW and TZ, respectively, compared with that in the SW. In sesquiterpenoid and triterpenoid biosynthesis, we determined that the relative contents of two sesquiterpenoids (guaiadiene and cubebene) in HW and TZ were higher than those in SW. Our transcriptomic and metabolomic data show that terpenoid biosynthesis takes place in situ in the HW using precursors (starch and sugars) that are produced by primary metabolism.

### 2.5. Transcription Factors Related to Terpenoid Synthesis

There were 14 differentially expressed transcription factors (TFs) identified in the DEGs. The differentially expressed TFs were annotated to encode AP2/ERF-ERF, bHLH, C2H2, GARP-G2-like, MYB, MYB-related, NAC, OFP, SBP, and WRKY (Figure 6A). The MYB, MYB-related, NAC, OFP, and bHLH TF unigenes were upregulated in SW compared with those in HW and TZ. The Maker00030001.gene in AP2/ERF-ERF, Maker0050290.gene in C2H2, Maker00010783.gene in SBP, and Maker00039996.gene in WRKY were upregulated in HW. We further performed RT-qPCR to verify the correctness of the differential TFs identified by RNA-seq. The ten TF families exhibited the same trends in RT-qPCR and FPKM (Appendix A). Next, we attempted to identify TFs related to terpenoid synthesis through a correlation network analysis. The results showed that 11 TF unigenes (two MYB, two C2H2, and one MYB-related, NAC, OFP, SBP, WRKY, AP2/ERF-ERF, and GARP-G2-like each) displayed significant and higher correlations with terpenoid metabolites (Figure 6B). One MYB-related (Maker00003918.gene) and one AP2/ERF-ERF (Maker00010133.gene) TF unigene showed a strong negative correlation with the four terpenoid metabolites identified in this study. In contrast, four TF unigenes (one SBP, Maker00010783.gene; one WRKY, Maker00039996.gene; one C2H2, Maker00050290.gene; and one GARP-G2-like, Phoebe bournei_newGene_5197) showed strong positive correlations with the four terpenoid metabolites.

### 2.6. Phenylpropanoid and Flavonoid Biosynthesis Pathways

In Figure 7, two unigenes (one CAD gene, Phoebe_bournei_newGene_11548 and one COMT gene, Maker00033335.gene) in the phenylpropanoid biosynthesis pathways were upregulated in HW, and the other two unigenes were downregulated, which contributed to the synthesis of various lignins in HW. In addition, two unigenes (one PGT1 gene, Maker00031067.gene, and one ANS gene, Maker00027670.gene) in flavonoid pathways were downregulated in HW. Since the flavonoid pathway was upstream of anthocyanin biosynthesis, the downregulated anthocyanidin synthase (ANS) gene in HW vs. SW decreased the amount of substrate for anthocyanin metabolites.

## 3. Discussion

The heartwood (HW) of *Phoebe hui* is of particular economic value due to its special fragrance and golden-thread color. The polyphenoloxidase (PPO) and peroxidase (POD) activities were observed in differentiating xylem close to the cambium [10]. The PPO activity in *P. hui* sapwood was significantly higher than that in HW, which may imply its involvement in lignification, as has been shown for different laccases [30]. Higher POD activity was observed in the transition zone, and it is postulated that heartwood formation begins in the transition zone, which supports the view of previous studies that the transition zone is generally characterized by enhanced physiological activity [10,31]. Enzyme activities were needed for the biosynthesis of heartwood substances as well as the storage materials. In *P. hui*, the stored starch provided the primary metabolites needed for extractive formation. The phenol, flavonoid, and lipid contents increased gradually from the SW to the HW along with the gradual decrease in starch, and the deposition of phenols, flavonoids, and lipids in the HW would be closely related to the golden-thread color of the wood [1,13,25]. A similar phenomenon has long been observed in trees, where the disappearance of starch is correlated with the accumulation of extractives in heartwood [6,32,33].

From previous studies, it is well known that the fragrances in most woods, flowers, fruits, and leaves are volatile metabolites such as terpenoids, esters, terpene alcohols, and volatile benzenoid-phenylpropanoids [8,32,33,34,35]. Terpenoids are a class of natural organic compounds composed of isoprene structural units widely found in nature [36]. The special flavor of many plants is a result of the presence of terpenoids [37]. In this study, 49 volatile metabolites were identified, of which monoterpenoids, sesquiterpenoids, and isoxazoles had significantly increased accumulations in HW. These results confirm the observations that the main chemical constituents of the essential oils of Lauraceae species are cubene, 3,7-guaiadiene, and torreyol [5,20]. These results indicate that the fragrance in heartwood mainly results from the accumulation of these terpenoids. Previous studies of the heartwood of *Cunninghamia fortunei* isolated a variety of monoterpenoids, sesquiterpenoids, and diterpenoids, which are present in large quantities in essential oil [16,38,39,40]. Moniodis et al. [27] reported that the fragrant heartwood oil of *Santalum spicatum* was a mixture of sesquiterpene olefins and alcohols. Several reports have also shown that sesquiterpenoids are the dominant metabolites of fragrance in *Phoebe* plants [20,21,22]. In this study, a significantly increased accumulation of monoterpenoids (limonene and p-cymene) and sesquiterpenes (cubebene and guaiadiene) in HW was revealed to be closely related to fragrance. It is reasonable to assume that there is terpenoid diversity in *P. hui* heartwood. Three models (Type-I: *Robinia*, Type-II: *Juglans*, and Type-III: *Santalum*) of heartwood formation have previously been described by Celedon and Bohlmann [1] based on the mode of secondary metabolite biosynthesis and deposition. The Type-III (*Santalum*) model represents trees with a terpene-rich HW, where terpene biosynthesis could not be explained based on resin ducts that were active in the SW [1]. As evidence for this model, the transcriptome and metabolites of *Santalum album* revealed that the sandalwood oil was extracted from mature heartwood, and transcripts of sesquiterpenoid biosynthesis, such as the mevalonic acid pathway, farnesyl diphosphate synthase, and TPS transcripts, were preferentially and specifically expressed in HW [15]. Recent works in *Callitropsis nootkatensis* also confirmed this review of the *Santalum*-type model, which identified the HW-specific expression of genes encoding the sesquiterpene synthase and valencene synthase [41,42]. In this study, *P. hui* contained heartwood-specific extractable oil, which is comprised mostly of terpene molecules. The accumulation of these compounds suggests active terpene biosynthesis during HW development. Jones et al. [43] suggested that ray parenchyma cells may be involved in sesquiterpene biosynthesis and accumulation in the HW. In this study, the complete RNA was isolated from HW, thereby indicating active cells present in the HW. Collectively, we speculated that the model of heartwood formation of *P. hui* could be classified as Type-III (*Santalum*) based on the biosynthesis and accumulation of monoterpenoids and sesquiterpenoids in the heartwood.

A better understanding of the regulated mechanisms by which these compounds are produced can help us to select and increase the production of desired compounds. During HW formation, reserves of SW are consumed [44]. Primary metabolism is extremely important for the biosynthesis of terpenoids because it provides the precursors for the biosynthesis of terpenoids. Herein, upregulated transcripts in pentose and glucuronate intercoversions, amino sugar and nucleotide sugar metabolism, and starch and sucrose metabolism were observed in SW, thereby suggesting sufficient precursors (starch and sugars) for the downstream synthesis pathways of terpenoid backbone biosynthesis and ultimately for the terpenoid biosynthesis pathways. Terpenoids in plants are produced by terpene synthases (TPSs) from linear prenyl diphosphates: geranyl diphosphate (GPP) for monoterpenes, farnesyl diphosphate (FPP) for sesquiterpenes, and geranylgeranyl diphosphate (GGPP) for diterpenes [26]. Momilactones, which are special diterpenoid metabolites, include the two major structurally divergent forms of momilactronones A and B [45]. Momilactone-A synthase (MAS) is an enzyme that catalyzes the production of momilactone A from geranylgeranyl-PP (GGPP) [46]. Upregulation of MAS genes may indicate a relatively high accumulation of momilactone A for monoterpenes in the HW. In the monoterpenoid biosynthesis pathway, (-)-terpineol synthase, a member of the TPS-d subfamily, is a predominant synthase that catalyzes the biosynthesis of terpineol [47,48]. We found that the (-)-terpineol synthase-related unigenes were downregulated in HW, indicating that the main monoterpenoid was limonene rather than terpineol—given the increased accumulation of limonene in HW. These findings (volatile metabolites and transcriptomic) indicate that the diversity of terpenoids in *P. hui* heartwood could be a mixture of monoterpenoids, diterpenoids, and sesquiterpenoids. In addition, the most apparent function of lignin is providing mechanical strength and rigidity to the cell wall and facilitating the formation of xylem vessels to transport water and nutrients over a long distance [49]. Due to its contribution to plant development, lignin forms a barrier against microbial infections and pest herbivory, serving as one of the major contributors to biotic and abiotic stress resistance [50,51,52]. In this study, the expression of two phenylpropanoid biosynthesis unigenes (PhCAD, Phoebe_bournei_newGene_11548 and PhF5H, Phoebe_bournei_newGene_2299) in HW was higher than those in SW. The high expression of these genes leads to sufficient accumulation of precursor compounds for the synthesis of lignin in HW. The accumulation of lignin (e.g., p-hydroxyphenyl lignin, guaiacyl lignin, 5-hydroxy guaiacyl lignin, and syringyl lignin) in heartwood contributes to its higher density, durability, and stress resistance compared with sapwood.

Transcription factors (TFs) are involved in the regulation of secondary metabolite biosynthesis by activating or inhibiting the expression of genes [53,54,55]. To date, TF families have been reported in plants that are involved in the biosynthesis of terpenoids and phenolics [56]. In many tree plants, MYB and WRKY were identified as the key TFs that regulated the production of terpenoids [43,57,58]. In *P. hui*, two MYB unigenes showed a significant negative correlation with the contents of limonene and p-cymene, which suggests that the MYB TFs would be a negative regulator of monoterpenoid biosynthesis through the suppressed expression of downstream target genes. WRKY TFs also emerged as a key family in terpene biosynthesis [59]. Some WRKY TFs transactivate the promoters of TPSs to regulate terpene biosynthesis [57,58,59,60,61]. One WRKY unigene (Maker00039996.gene) was upregulated in the heartwood, which promoted the production of monoterpenoids and sesquiterpenoids in *P. hui*. Study of the regulatory role of C2H2 on terpenoid synthesis has been limited, but Xiang et al. [62] and Sharma et al. [63] reported enhancements of terpenoid, sesquiterpenoid, and triterpenoid biosynthesis together with the upregulation of genes in the C2H2 family. Therefore, the present results indicate that the synthesis of monoterpenoids and sesquiterpenoids during heartwood formation was probably the outcome of C2H2 expression (Maker00050290.gene). Nieuwenhuizen et al. [64] reported that NAC could activate AcTPS1 to regulate monoterpenoid synthesis in *Actinidia chinensis*. CitERF71 activated the promoter of CitTPS16, which led to the biosynthesis of geraniol in *Citrus sinensis* [65]. Interestingly, the negative regulation of terpenoid biosynthesis by NAC and AP2/ERF-ERF was in contrast to the results of previous studies. This implies potential novel NAC and AP2/ERF-ERF TFs in *P. hui*. We will further verify these transcription factors and provide a more accurate theoretical basis for the breeding of *P. hui* and the regulation of terpenoids.

## 4. Materials and Methods

### 4.1. Plant Material and Sampling

The *Phoebe hui* trees were planted in a nursery (30°71′ N, 103°77′ E) in Chengdu, Sichuan Province, China. The diameter of the trunk at breast height (1.3 m) of three *P. hui* trees (65 years old) were sampled at 9:00–11:00 a.m. in July 2021 after cutting down the trees with a chain saw. One 5 cm thick basal wood disk per tree was then divided into a strip and carefully separated into three tissues along the radial section according to the boundaries defined by color. The inner wood with dark color was identified as heartwood (HW), the outer wood with light color was identified as sapwood (SW), and the narrow zone (approximately three annual rings) of the intermediate developmental stage between SW and HW was identified as the transition zone (TZ) (Figure 8). Each fresh tissue sample was then immediately divided into two parts. One part was immediately frozen in liquid nitrogen and stored at −80 ℃ for transcriptome sequencing, metabolite extraction, and physiological testing. The other part was fixed and vacuumed with 4% paraformaldehyde for subsequent immunohistochemical staining analysis.

### 4.2. Microscopy Analyses and Physiological Tests

Radial sections (30 μm in thickness) from each tissue were obtained at −20 ℃ using a cryostat (Leica CM1900). To observe the starch grains and lipids, radial sections from the three tissues were stained with I2-KI and Sudan IV, respectively. After the staining process, the sections were gradually dehydrated and cleared with 30% (5 min), 50% (5 min), 70% (5 min), 90% (5 min), 95% (5 min), and 100% alcohol (10 min) and alcohol:dimethylbenzene (*v*:*v* = 1:1) (2 min) and dimethylbenzene (4 min). The sections were then mounted on microscopic slides and observed under an optical microscope (Olympus BX 50). The flavonoid and total phenol contents and the enzyme activities of peroxidase (POD) and polyphenol oxidase (PPO) in SW, TZ, and HW were measured using test kits for plant flavonoid, total phenol, POD, and PPO (Nanjing Jiancheng Bioengineering Institute, Nanjing, China), respectively. A one-way ANOVA (*p* < 0.05) followed by the LSD multiple comparisons test was used to compare differences among the three tissues.

### 4.3. GC-MS Analysis

Two grams of sample per tissue was ground into powder using a tissue grinder (Kunshan, China) with glass beads (Sigma-Aldrich, Shanghai, China) for 90 s at 60 Hz. The metabolites were extracted using the Soxhlet extraction method with 1 mL ethyl acetate and put into an injection bottle for gas chromatography-mass spectrometry (GC-MS). The extracts were analyzed using an Agilent GC8890 equipped with MS5977 (Agilent, Shanghai, China). The employed temperature program had the following settings: an initial temperature of 60 °C, increased to 160 °C at 8 °C/min, and then increased to 220 °C at 10 °C/min. The total GC runtime was 18.5 min. Nitrogen was used as the carrier gas at a flow rate of 1 mL/min. The ion source and quadrupole were 200 and 150 °C, respectively. The solvent delay time was 5.0 min. The ion energy for electron impact was 70 eV. The mass scan range was m/z 50–500. Identification of metabolites was performed using the public databases HMDB [66], Massbank [67], LipidMaps [68], and mzcloud [69]. The identification of metabolites was calculated using the retention time and fragmentation mode. The relative content of components was obtained from the ratio of the peak area of each metabolite to the total peak area. The filtering conditions for differentially accumulated metabolites (DAMs) were as follows: absolute log2 (fold change) ≥ 1, *p*-value < 0.05, and variable importance in projection (VIP) ≥ 1. To study the accumulation of specific metabolites, a principal component analysis (PCA) was performed using the ‘rplos’ package in R, and an orthogonal partial least squares-discriminant analysis (OPLS-DA) was performed using MetaboAnalyst 5.0 [70].

### 4.4. RNA-seq Analysis

Total RNA was extracted from frozen samples using the TaKaRa MiniBEST Plant RNA Extraction Kit (TaKaRa, Dalian, China) according to the manufacturer’s instructions. Nanodrop and 2100/GX were used to test the integrity of the total RNA. The mRNA library of each sample was constructed and sequenced on the Illumina HiSeq 4000 platform. The adaptor and low-quality sequences were removed using Fastp with default parameters [71] and clean reads were then mapped to the *Phoebe bournei* genome (CRR101285, https://ngdc.cncb.ac.cn/gsa/browse/CRA002192, accessed on 31 December 2020) [72] using HISAT2 [73]. FPKM was used for gene-/transcript-level quantification. Based on the raw count data, a differential expression analysis between samples was performed with DESeq2 software [74]. Genes with a |log2 (fold change)| > 1 and FDR (false discovery rate) < 0.05 were defined as differentially expressed genes (DEGs) and subjected to Gene Ontology (GO) (http://geneontology.org/, accessed on 31 December 2020, Copyright © 2022–2022 Member of the Open Biological Ontologies Foundry) and Kyoto Encyclopedia of Genes and Genomes (KEGG) (https://www.kegg.jp/, accessed on 31 December 2020, Copyright 1995–2022 Kanehisa Laboratories) [75] enrichment analyses. Enriched genes were further analyzed with gene set enrichment analysis (GSEA) [76].

Total RNA was isolated, and first-strand cDNA was synthesized using a PrimeScriptTM RT reagent kit with gDNA Eraser (TaKaRa, Dalian, China). Twenty pathway genes and transcription factors (TFs) related to terpenoid, phenylpropanoid, and flavonoid biosynthesis factors were selected for RT-qPCR analysis, and the actin gene was used as the internal control for the normalization of gene expression (Appendix A). RT-qPCR was performed using TB Green Premix Ex TaqTM II (TaKaRa, Dalian, China) on a CFX96 Real-Time System (BIO-RAD, USA). Each sample was analyzed in three technical replicates.

### 4.5. Correlation Analysis of Metabolites and Transcription Factors

The transcription factors (TFs) were subjected to an association analysis of differentially accumulated terpenoids. A correlation analysis was performed by calculating the Pearson correlation coefficient (PCC) between the metabolite contents and transcriptional changes, and the screening criterion was PCC ≤ −0.8 or PCC ≥ 0.8 (*p* < 0.05). Cytoscape version 3.10.0 (The Cytoscape Consortium, San Diego, CA, USA) was used to visualize the interaction networks between TFs and terpenoids.

## 5. Conclusions

The aim of this investigation was to assess the biochemical basis and regulatory mechanism of fragrance in *Phoebe hui* heartwood. The cytological, physiological, volatile-metabolomic, and transcriptomic characteristics were compared using three radial tissues of trunk wood. Monoterpenoids (limonene and p-cymene) and sesquiterpenoids (cubebene and guaiadiene) were the dominant volatile metabolites related to the fragrance in *P. hui* based on their increased accumulation in the heartwood. The total amount of phenols, flavonoids, and lipids increased gradually from sapwood to heartwood and were concomitant with a decrease in starch. The pathway genes in primary metabolism and terpenoid biosynthesis were promoted in sapwood and heartwood, respectively. Eleven transcription factors were involved in regulating terpenoid biosynthesis. These findings clearly indicate that the heartwood formation belongs to the Type-III (*Santalum*) model, and the terpenoids were synthesized and accumulated in the heartwood using the precursors. This is the first study on the volatile metabolites that determined the regulation of heartwood fragrance by differentially expressed genes in *P. hui* (one of the tree species of golden-thread nanmu). These findings have significant implications for understanding the accumulation of fragrance-related metabolites in *P. hui* heartwood and provide a theoretical basis for future breeding work on wood improvement. Further research on the functions of relevant genes is needed to determine the detailed regulatory mechanism of the formation of fragrance-related metabolites.

## Figures and Tables

**Figure 1 ijms-23-14044-f001:**
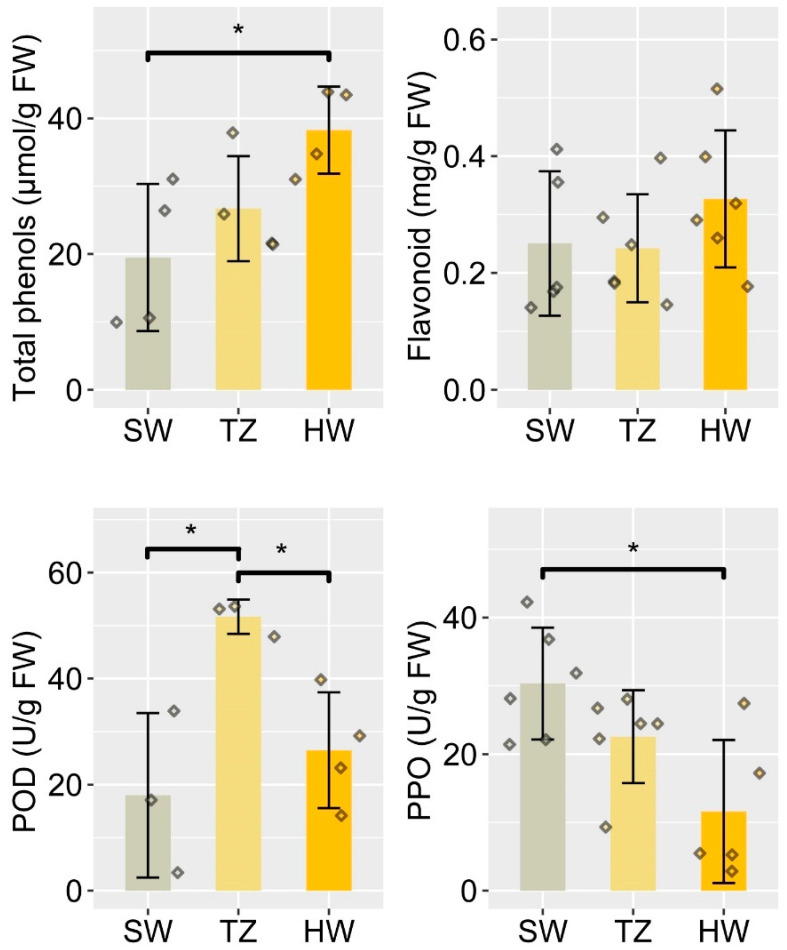
The contents of total phenols and flavonoids and the activities of POD and PPO in SW, TZ, and HW. * indicates a significant difference at the level of 0.05. The rhombus with different colors represented the biological replicates data of the contents of total phenols, flavonoids, and the activities of POD and PPO in different tissues.

**Figure 2 ijms-23-14044-f002:**
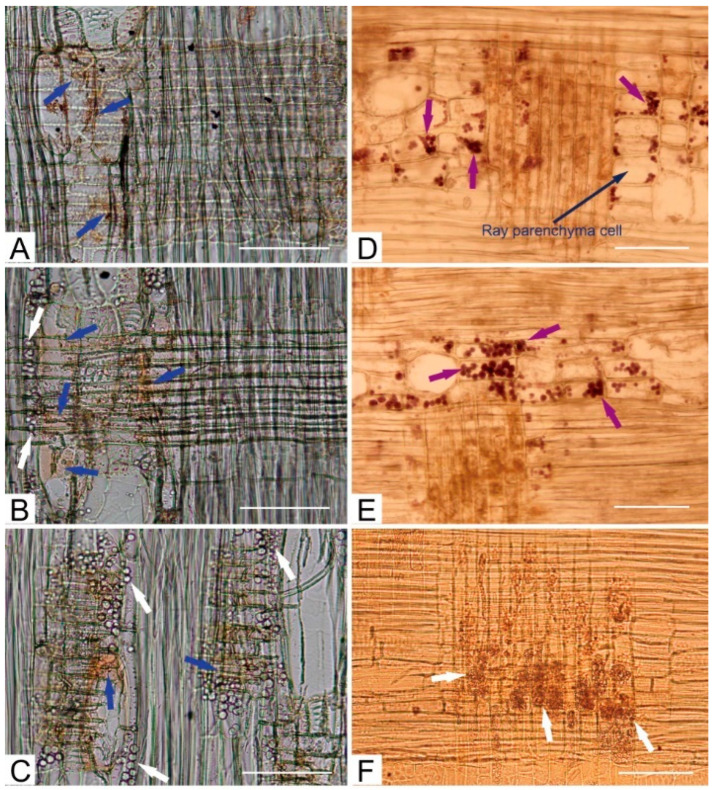
Light micrographs of starch grains, lipids, and extractives in *P. hui* radial sections. (**A**–**C**) SW, TZ, and HW sections stained with a Sudan IV solution showing lipids dyed orange-yellow (blue arrows) and extractives (white arrows). (**D**–**F**) SW, TZ, and HW sections stained with a I_2_-KI solution showing starch grains (purple arrows) and extractives (white arrows). Bar = 10 μm.

**Figure 3 ijms-23-14044-f003:**
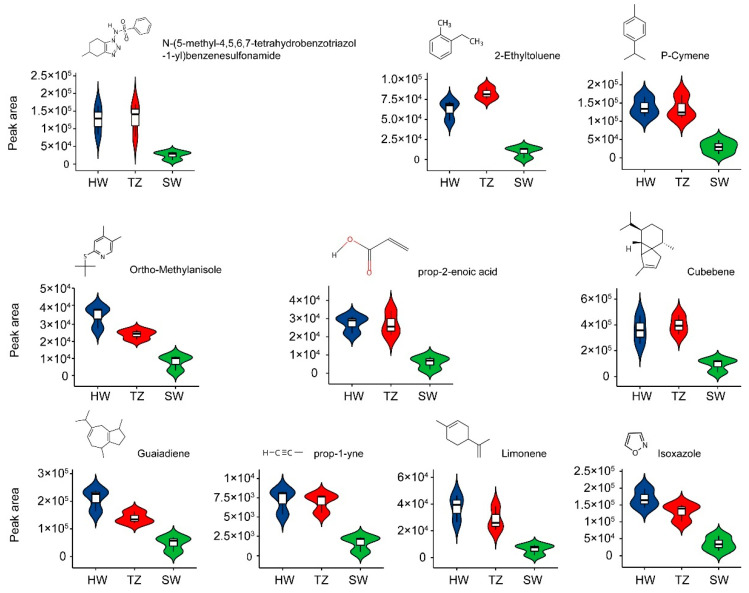
Structure and peak area of the top ten differential volatile metabolites related to fragrance.

**Figure 4 ijms-23-14044-f004:**
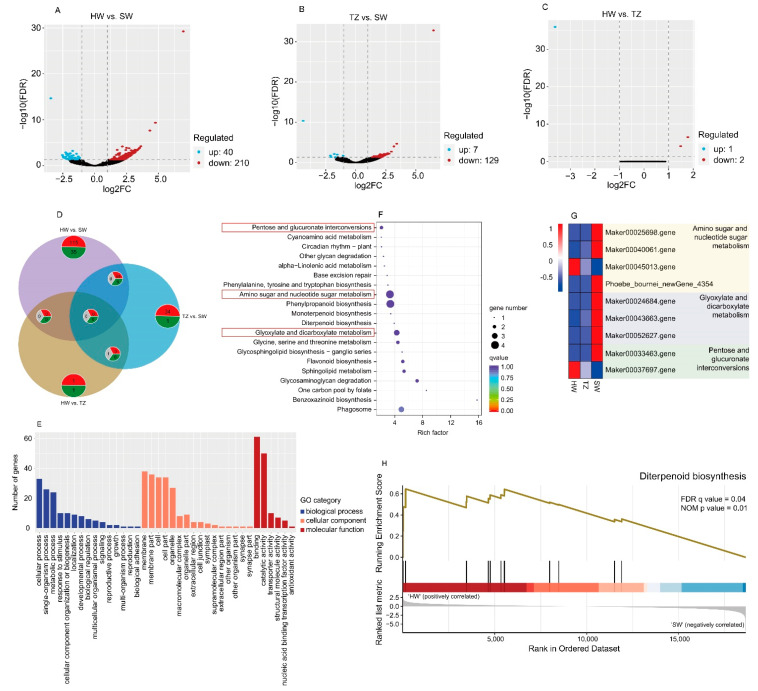
Preliminary analysis of transcriptomic data. (**A**–**C**) Volcano plots of differentially expressed genes (DEGs) for HW vs. SW (**A**), TZ vs. SW (**B**), and HW vs. TZ (**C**). (**D**) Venn diagram of DEGs. Red pie, number of downregulated genes; gray pie, number of upregulated genes; gray pie, number of contralateral genes in the two groups. (**E**) GO enrichment analysis of 150 DEGs exclusively detected in HW vs. SW. (**F**) Top 20 KEGG pathways with the most significant enrichment among the 150 DEGs. (**G**) Heatmap of genes related to amino sugar and nucleotide sugar metabolism, glyoxylate and dicarboxylate metabolism, and phenylpropanoid biosynthesis among HW, TZ, and SW from RNA-seq. (**H**) GSEA (gene set enrichment analysis) of genes related to diterpenoid biosynthesis. The GSEA rank was calculated for HW vs. SW.

**Figure 5 ijms-23-14044-f005:**
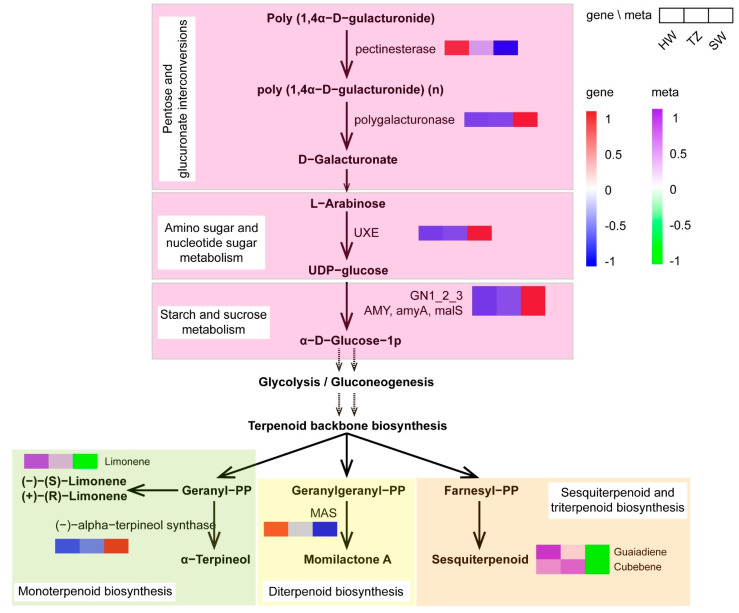
Schematic diagram of primary and secondary metabolism pathways (terpenoid biosynthesis) activated in the HW. The pathways with pink boxes represent the primary metabolic pathway. UXE: UDP-arabinose 4-epimerase. GN1_2_3: glucan endo-1,3-beta-glucosidase 1/2/3, AMY, amyA. malS: alpha-amylase. MAS: momilactone-A synthase.

**Figure 6 ijms-23-14044-f006:**
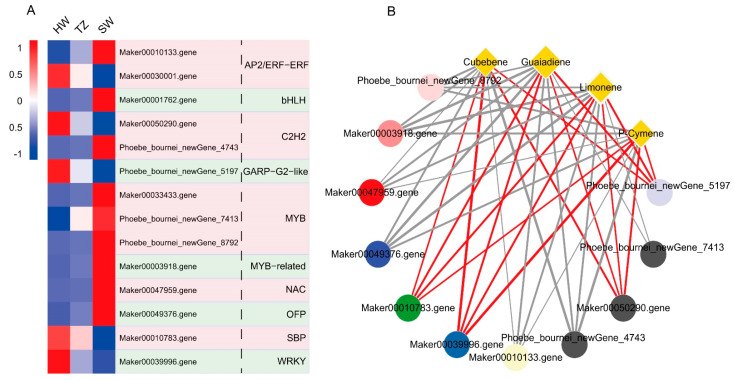
(**A**) Differentially expressed transcription factors among HW, TZ, and SW and (**B**) connection network between TFs and terpenoid metabolites. The yellow diamonds represent the terpenoid metabolites, the thickness of the diamond frame represents the fold change (FC) of terpenoid metabolites between HW and SW, and the size of the diamond represents the number of genes related to terpenoid metabolites. A total of 10, 11, 11, and 10 TFs showed significant correlation with the content of P-Cymene, Limonene, Guaiadiene, and Cubebene, respectively. The red/gray lines show positive/negative correlations, respectively, between terpenoid metabolites and TFs, and the thickness of the line represents the value of the correlation coefficient (thick to thin: low to high coefficient value). The circles represent TF unigenes, and the colors of the circles represent TF families.

**Figure 7 ijms-23-14044-f007:**
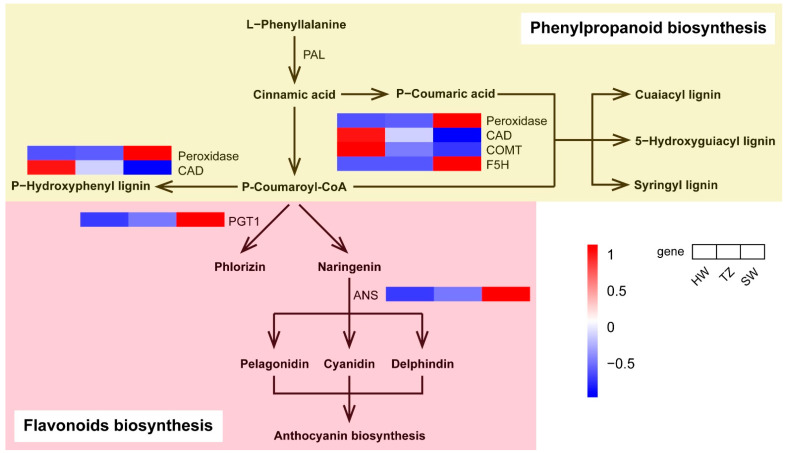
Transcript profiling of genes in the phenylpropanoid and flavonoid biosynthesis pathways among HW, TZ, and SW. PAL, phenylalanine ammonia-lyase; CAD, cinnamyl-alcohol dehydrogenase; COMT, caffeic acid 3-O-methyltransferase/acetylserotonin O-methyltransferase; F5H, ferulate-5-hydroxylase; PGT1, phlorizin synthase; and ANS, anthocyanidin synthase.

**Figure 8 ijms-23-14044-f008:**
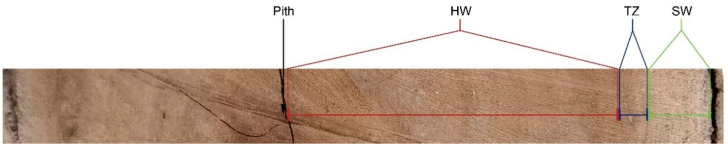
The sampling locations of different tissues of the *P. hui* wood strips. SW, sapwood; TZ, transition zone; and HW, heartwood.

## Data Availability

The datasets presented in this study can be found in online repositories. The names of the repository/repositories and accession number(s) can be found below: PRJNA881959 (https://www.ncbi.nlm.nih.gov/bioproject/PRJNA881959, accessed 19 September 2022).

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
