# Peer review of "Integrated Transcriptomic, Metabolomic, and Physiological Analyses Reveal New Insights into Fragrance Formation in the Heartwood of Phoebe hui"

_ijms, 2022, doi:10.3390/ijms232214044_

Round 1

Author Response

Dear reviewer,

Thank you for your comments concerning our manuscript entitled “Integrated transcriptomic, metabolomic and physiological analysis reveals new insights into fragrance formation in heartwood of Phoebe hui” (ijms-1969061). Those comments are all valuable and very helpful for revising and improving our paper, as well as the important guiding significance to our researches. We have studied the comments carefully and have made correction which we hop meet with approval. Revised portion are marked in red in the paper. The main corrections in the paper and the responds to your comments are as following:

  1. The text has lots of problems with its English. Overall, about one in every three sentences has a mistake in its English. I will not point out all of these, because it would take up more time than I have available.  I will give a few examples, however. 

Response to comment: It is really true as you suggested that the text has lots of problems with English. Therefore, the revised manuscript has been polished by a professional academic polishing company (AJE, American Journal Experts).

  1. The sentence on lines 34-35 has no verb.

Response to comment:  We have made correction according to your comment. Please review line 34-35.

  1. On line 44, there is no such thing as “almost traditional”.

Response to comment: Considering your suggestion, we have revised “almost traditional …” as “… the society”. Please review line 45.

  1. On line 100, I have no idea what “the heartwood-specially pattern” might mean.

Response to comment: This sentence means that the biosynthesis of terpenoid happened at heartwood. We have made correction in the text. Please review line 99-100.

  1. No idea what “still majors were present” (line 128) might mean.

Response to comment: This sentence means that the starch mainly accumulated in SW and TZ. We have made correction in the text. Please review line 128-131.

  1. Lines 167,168 are not a sentence. “be to be” is not English (line 181). Line 191, “that enriched” should be “that were enriched”. I am guessing that “is particularly economical valued” (line 185) is supposed to mean “is of particular economic value”.

Response to comment: We have made correction according to your comment. Please review line 167-168, 181-182, and 290.

  1. No idea what “small, but complete RNA” might mean (lines 340, 341).

Response to comment: This sentence means that the complete RNA could isolate from the HW. We have made correction in the text. Please review 338.

  1. Some words are strange or undefined. Is a “lipide” supposed to mean a lipid?  They do use the word lipid a few times, so I am wondering what is different about a “lipide”.

Response to comment: “lipide” mean a lipid. We are very sorry for our incorrect writing and we have made correction in the text. Please review line 128, 138, and 428.

  1. What are “DAMs” in their manuscript? If they defined the term, I missed the definition.

Response to comment:  DAMs mean differential accumulated metabolites. We have supplied the definition in the location of first appearance in the text. Please review line 155.

  1. Line 126: Mentions “ray cells”, but most readers will not know what these are.

Response to comment: We have marked the ray parenchyma cell in Figure 2. Please review the revised Figure 2.

  1. Lines 117 and 139,140 talk about results without a description of the experiments that yielded the results. At least provide a sentence or two saying what was done.  The details can be left in the Methods but the general experiment needs to be mentioned in the Results.

Response to comment: Considering your suggestion, we have supplied the description of experiments of physiological and volatile metabolites test that yielded the results in the section of results. Please review line 117-121, and 143-144.

  1. Figure 3 is too complicated and is also not essential. Better to move it to Supplementary Materials, although 3A is something worth retaining.  I doubt that many scientists know what an “Upset diagram of DAMS” might be, so some explanation would be useful.

Response to comment: As you suggested that we have moved Figure 3 to supplementary figures. And we also explained the “upset diagram of DAMs” in the section of figure title. Please review the revised Figure S2.

  1. Lines 214, 215 make the statement that “Primary metabolism provided (sic, should be provides) the precursors to the biosynthesis of secondary metabolites”. This is true, of course, but that is the definition of secondary metabolites.  That is, this story is true for every secondary metabolite, and thus not some amazing discovery of this study.  Lines 348-350 are super silly, because the vital importance of primary metabolism for every secondary product on the planet is well known, so the authors should tone this down.

Response to comment: It is really true as you suggested that primary metabolism provides the precursors to the biosynthesis for every secondary metabolites. And we have made correction according to your comments. Please review line 213, and 339-340.

  1. Authors repeatedly say “On the contrary” (no such English phrase exists) when they mean “In contrast”.

Response to comment: We have made correction according to your comment. Please review line 218 and 230.

  1. Figure 5 is very minor, but could be retained in Supplementary info.

Response to comment: Figure 5 have moved to supplementary figures. Please review Figure S3.

  1. Because the authors make a big deal of this, it would be good if the Legend and coloring on Figure 7 indicated which of the steps are “primary metabolism”. As a plant biochemist, I know this, but many readers will not.

Response to comment: Considering your suggestion, we revised the legend and coloring the primary metabolism in Figure 5. And we also supplied notes of primary metabolism of the figure in the section of figure title. Please review the revised Figure 5.

  1. As with all of their work, the transcriptomic analysis yields correlations, not causality. There are several places in the Discussion where causality conclusions are made, and they are all unacceptable.  The most severe is on lines 520-521.

Response to comment: We have made correction in the section of discussion and conclusion according to your comment. Please review line 310-312, 318-322, 493-495, etc.

  1. Some of these eleven transcription factors may have nothing to do with terpenoid biosynthesis. Lots of other things (e.g., lignin) are going on in the SP to TZ to HW transition, so how can they conclude that all correlated TF changes are about terpenoids?

Response to comment: The expression level of these eleven transcription factors showed significantly correlation with the content of terpenoid. The transcription factors would regulate the biosynthesis of terpenoid through regulated the expression of enzyme genes in the terpenoid biosynthesis pathway. While, the regulation mechanism of transcription factors in terpenoid biosynthesis should be further verify.

  1. The Discussion is way too long for the modest discoveries made. I would suggest reducing it about 2-fold so as to not bore the reader too much.

Response to comment: We have reduced the length of discussion. Please review the section of discussion.

  1. The references are very complete, so the authors should be praised for citing the relevant literature very well. However, the references lack italics for the genus and species names that are in many of the manuscripts that they cite.

Response to comment: We are sorry for our negligence. We have revised the italics of the genus and species names in the reference. 

  1. The authors state “boldly” for rather obvious actions they take. If they consider these bold, then these folks need some more excitement in their lives.

Response to comment: We are sorry for our incorrect writing, and we have made correction in the text. Please review line 33 and 338.

Special thanks to you for your good comments.

We appreciate for your warm work earnestly, and hope that the correction will meet with approval.

Reviewer 2 Report

The authors combined transcriptomic and metabolomic analysis to reveal the key fragrance molecules and related genes in the bio-generation pathways. The findings were also validated by RT-qPCR and phenotypic demonstrations. The story is interesting, and the paper is well written with appropriate methodology, precisely described results and adequate discussion. I have minor comments:

1.      Section 2.1, The authors should briefly describe the relationship between PPO, POD, starch and lipide and the purpose of the study herein.

2.      Lines 142 and 147, PCA, OPLS-DA: Abbreviations should be given the full name for the first time.

3.      Line148, what are the digitals 24, 23 and 27 stand for?

4.      Line 163, in the caption of Figure 4, ten significant ..., does it mean 'top ten ....'? make it clear.

5.      Line 170, What do these numbers 18, 021, 025-24,142, 387 refer to? Please make it clear.

6.      Figure 6E, the font size of the axis scales is too small to see.

7.      Line 221, Lowercase “uxe” is redundant.

8.      Figure 8: I strongly recommend the authors re-plot Figure 8B using cytoscape or the R package of igraph. It looks not so good. The authors should give a legend showing the relationship between node size and corresponding number.

9.      Line 464, The abbreviation HPLC has nothing to do with ethyl acetate. Please delete HPLC. How much solvent was used in extraction? If the sample was diluted before feeding, the author also needs to give a description

10.    Line 473, I have never known that KEGG provides a reference database for MS. I think KEGG doesn't have this function. Please check it.

11.    Line 473, "The quantification...", should be "The identification...".

12.    Line 528, What does wood414 refer to?

Author Response

Dear reviewer,

Thank you for your comments concerning our manuscript entitled “Integrated transcriptomic, metabolomic and physiological analysis reveals new insights into fragrance formation in heartwood of Phoebe hui” (ijms-1969061). Those comments are all valuable and very helpful for revising and improving our paper, as well as the important guiding significance to our researches. We have studied the comments carefully and have made correction which we hop meet with approval. Revised portion are marked in red in the paper. The main corrections in the paper and the responds to your comments are as following:

  1. Section 2.1, The authors should briefly describe the relationship between PPO, POD, starch and lipide and the purpose of the study herein.

Response to comment: Considering your suggestion, we have supplied the relationship and purpose of those physiological indexes in our study. Please review line 117-121.

  1. Lines 142 and 147, PCA, OPLS-DA: Abbreviations should be given the full name for the first time.

Response to comment: We have made correction according to your comment. Please review line 148, 153.

  1. Line148, what are the digitals 24, 23 and 27 stand for?

Response to comment: The digitals 24, 23, and 27 stand for differential accumulated metabolites (DAMs) was screened from the comparison group of HW vs. SW, HW vs. TZ, and TZ vs. SW, respectively. We have made correction in the text. Please review line 155-156.

  1. Line 163, in the caption of Figure 4, ten significant ..., does it mean 'top ten ....'? make it clear.

Response to comment: We have made correction according to your comment. Please review line 165.

  1. Line 170, What do these numbers 18, 021, 025-24,142, 387 refer to? Please make it clear.

Response to comment: These numbers 8, 021, 025-24,142, 387 refer to the number of clean reads. We have made correction in the text. Please review line 170.

  1. Figure 6E, the font size of the axis scales is too small to see.

Response to comment: We have revised the font size of the axis scales in Figure6E. Please review the revised Figure 4E.

  1. Line 221, Lowercase “uxe” is redundant.

Response to comment: We have deleted the lowercase of “uxe”. Please review line 221, 241.

  1. Figure 8: I strongly recommend the authors re-plot Figure 8B using cytoscape or the R package of igraph. It looks not so good. The authors should give a legend showing the relationship between node size and corresponding number.

Response to comment: We have re-plot this figure using cytoscape. And we also supplied a legend showing the relationship between node size and corresponding number after the title of figure. Please review the revised Figure 6B and line 267-269.

  1. Line 464, The abbreviation HPLC has nothing to do with ethyl acetate. Please delete HPLC. How much solvent was used in extraction? If the sample was diluted before feeding, the author also needs to give a description

Response to comment: We have deleted HPLC. 1mL ethyl acetate was used in extraction, and the sample was not diluted before feeding. We have made correction in the text. Please review line 433.

  1. Line 473, I have never known that KEGG provides a reference database for MS. I think KEGG doesn't have this function. Please check it.

Response to comment: We are very sorry for our incorrect writing. And we have deleted it. Please review line 442.

  1. Line 473, "The quantification...", should be "The identification...".

Response to comment: We have made correction according to your comment. Please review line 442.

  1. Line 528, What does wood414 refer to?

Response to comment: We are very sorry for our negligence of the “wood414”. This is a mistake. “wood414 improvement” refer to “wood improvement”. We have made correction in the text. Please review line 497.

Once again, thank you very much for your comments and suggestions.